# O-Band Emitting InAs Quantum Dots Grown by MOCVD on a 300 mm Ge-Buffered Si (001) Substrate

**DOI:** 10.3390/nano10122450

**Published:** 2020-12-07

**Authors:** Oumaima Abouzaid, Hussein Mehdi, Mickael Martin, Jérémy Moeyaert, Bassem Salem, Sylvain David, Abdelkader Souifi, Nicolas Chauvin, Jean-Michel Hartmann, Bouraoui Ilahi, Denis Morris, Ali Ahaitouf, Abdelaziz Ahaitouf, Thierry Baron

**Affiliations:** 1Univ. Grenoble Alpes, CNRS, CEA-Leti, Grenoble INP, LTM, F-38054 Grenoble, France; abouzaidoumaima@gmail.com (O.A.); hussein.mehdi@cea.fr (H.M.); mickael.martin@cea.fr (M.M.); Jeremy.moeyaert@cea.fr (J.M.); Sylvain.david@cea.fr (S.D.); 2SIGER Laboratory, Faculty of Sciences and Technology, Université Sidi Mohammed Ben Abdellah, Fès BP. 2202, Morocco; ali.ahaitouf@usmba.ac.ma; 3Institut des Nanotechnologies de Lyon (INL)-UMR5270-CNRS, Université de Lyon, INSA-Lyon, 7 avenue Jean Capelle, 69621 Villeurbanne, France; abdelkader.souifi@insa-lyon.fr (A.S.); nicolas.chauvin@insa-lyon.fr (N.C.); 4Univ. Grenoble Alpes, CEA-LETI, F-38054 Grenoble, France; jean-michel.hartmann@cea.fr; 5Institut Quantique et Laboratoire Nanotechnologies Nanosystèmes (LN2)-CNRS UMI-3463, 3IT, Université de Sherbrooke, Sherbrooke, QC J1K 2R1, Canada; bouraoui.ilahi@usherbrooke.ca (B.I.); Denis.Morris@USherbrooke.ca (D.M.); 6Faculté Polydisciplinaire Taza, Université Sidi Mohammed Ben Abdellah, LSI, Taza B.P. 1223, Morocco; abdelaziz.ahaitouf@usmba.ac.ma

**Keywords:** Quantum Dots (QDs), semiconductor III-V, Metal Organic Chemical Vapor Deposition (MOCVD)

## Abstract

The epitaxy of III-V semiconductors on silicon substrates remains challenging because of lattice parameter and material polarity differences. In this work, we report on the Metal Organic Chemical Vapor Deposition (MOCVD) and characterization of InAs/GaAs Quantum Dots (QDs) epitaxially grown on quasi-nominal 300 mm Ge/Si(001) and GaAs(001) substrates. QD properties were studied by Atomic Force Microscopy (AFM) and Photoluminescence (PL) spectroscopy. A wafer level µPL mapping of the entire 300 mm Ge/Si substrate shows the homogeneity of the three-stacked InAs QDs emitting at 1.30 ± 0.04 µm at room temperature. The correlation between PL spectroscopy and numerical modeling revealed, in accordance with transmission electron microscopy images, that buried QDs had a truncated pyramidal shape with base sides and heights around 29 and 4 nm, respectively. InAs QDs on Ge/Si substrate had the same shape as QDs on GaAs substrates, with a slightly increased size and reduced luminescence intensity. Our results suggest that 1.3 μm emitting InAs QDs quantum dots can be successfully grown on CMOS compatible Ge/Si substrates.

## 1. Introduction

III-V compounds have unique properties such as direct bandgap and high electron mobility, which are decisive for applications ranging from high frequency data transmission to light emission or photo-detection [1]. Combining these advantages with the maturity of the silicon CMOS industry opens the way towards Si photonic devices [2,3]. Bonding of III-V devices on silicon platforms is already a reality. A more prospective way could be the integration of III-V semiconductors directly on silicon wafers by heteroepitaxy [4]. This solution remains very challenging due to differences in polarity [5], lattice parameter [6], and thermal expansion coefficient between III-V and silicon [7], resulting in structural defects which are detrimental for electronic and optical properties. Those hurdles can be overcome by growing quantum dots instead of 2D epitaxial films [8]. Furthermore, high crystalline quality Ge layers on Si substrates help in obtaining superior quality III-V QDs on Si wafers [9]. In addition to their robustness against structural defects [8], InAs QDs offer widely tunable emission wavelengths that can be controlled in-situ [10] or ex-situ [11], offering the opportunity to integrate various III-V optoelectronic devices on Si substrates.

In this study, we carried out the growth as well as the structural and the optical characterization of InAs quantum dots on 300 mm quasi-nominal Si(001) substrates covered by Ge/GaAs bilayer buffers. We also combined optical emission at 1.3 µm on silicon substrates and numerical modeling to predict the QDs’ typical size and shape. A µPL mapping on the whole 300 Si substrate was carried out on more than 600 points. This study is an important step towards the integration of III-V semiconductors on 300 mm diameter Si substrates used in mainstream CMOS fabrication lines.

## 2. Experiments

The III-V epitaxy of InAs/GaAs QDs (Figure 1) was performed in an Applied Materials Metal Organic Chemical Vapor Deposition (MOCVD) reactor (Santa Clara, California, United States of America) on 300 mm Si(001) substrates with an offcut angle below 0.5° towards one of the crystallographic <110> directions. Those Si starting wafers are fully compatible with the silicon manufacturing industry.

First, 1.5 µm thick Ge Strain-Relaxed Buffers (SRBs) were grown in an Epi Centura Reduced Pressure-Chemical Vapor Deposition (RP-CVD) industrial cluster tool from Applied Materials (Santa Clara, California United States of America). Germane (GeH_4_) diluted in H_2_ was used as a Ge source. A low temperature/high temperature approach (more precisely, 400 °C, 100 Torr/750 °C, 20 Torr) was used, together with some short duration thermal cycling between 750 and 875 °C, to minimize the Threading Dislocations Density (TDD) [12,13]. With this growth method, we obtained smooth, cross-hatched Ge buffers with a surface root mean square roughness (RMS) around 0.8 nm for 20 µm × 20 µm Atomic Force Microscopy (AFM) (Figure 2a) images and a Threading Dislocation Density (TDD) around 10^7^ cm^−2^ [9]. Prior to GaAs heteroepitaxy, a Siconi^TM^ treatment followed by an annealing at 700 °C under hydrogen was used to remove native oxide on the Ge SRBs and prepare the surface [8]. After that, Ge/Si wafers were loaded in the MOCVD chamber equipped with Trimethylgallium (TMGa), tertiarybutylarsine (TBAs) and trimethylindium (TMIn) organometallic precursors (Ga, As, and In sources, respectively).

A 350 nm thick GaAs buffer was grown at a temperature of 620 °C and a pressure between 20 and 100 Torr, with no Anti-Phase Boundaries (APBs) as shown in AFM with a RMS roughness of 1.09 nm (Figure 2b) [14]. Then, InAs QDs were grown at a temperature of 520 °C for 40 s. The growth was then interrupted for 20 s under TBAs. Their growth rate was estimated at 0.18 Å/s, as we have no in-situ control of the amount of deposited indium. The QDs laid on top of a 2 nm thick In_0.12_Ga_0.88_As underlying layer. A 30 nm GaAs cap was added at 520 °C to complete the heterostructure. Using Ge SRBs, with a lattice parameter close to that of GaAs (a_Ge_ = 5.658 Å ⇔ a_GaAs_ = 5.653 Å), between the Si substrate and the GaAs buffer, significantly reduced the TDD, typically from 10^9^ cm^−2^ to 10^7^ cm^−2^, which should improve the optical properties of InAs/GaAs QDs. Reference samples were grown on GaAs(001) substrates with the same growth conditions.

Photoluminescence measurements were performed at 300 K using a 532 nm diode-pumped solid-state laser with an excitation power density of 273 W/cm^2^. Micro-photoluminescence measurements were performed at 300 K using a 780 nm diode-pumped solid-state laser with an excitation power density between 100 W/cm^2^ and 21 kW/cm^2^. Then, the PL emission was dispersed by a spectrometer and detected by liquid nitrogen cooled InGaAs (IGA) photodetector. Micro-PL (µPL) enabled us to achieve higher power densities for the same nominal laser power (around 4 µm spot size in µPL against around 200 µm in PL). Finally, tapping-mode AFM measurements were carried out on a Bruckers FastScanTM equipment.

## 3. Results and Discussion

Figure 3a–c shows 1 µm × 1 µm AFM images of uncapped InAs QDs from the center, the mid-radius, and the edge of the wafer, respectively. The QDs density is rather uniform over the whole 300 mm wafer surface, ranging from 6 × 10^9^ cm^−2^ at the center and mid-radius to 5 × 10^9^ cm^−2^ at the wafer edge. The average QD’s height is around 5 ± 0.11 nm, while the diameter is close to 40 ± 3 nm for the biggest ones.

However, it is well known that the capping process changes the QDs’ shape and size [15]. For that reason, cross-sectional TEM images of InAs QDs capped with 30 nm of GaAs are presented in Figure 4. QDs have a truncated pyramidal shape, a base length of 30 nm and a height of 5 nm. Eisele et al. [15] explained that the truncated pyramidal structure was due to diffusion during the capping.

To investigate the optical properties of these QDs, PL measurements at 300 K were performed at three different positions: the center, mid-radius, and edge of the wafer (Figure 5). Three peaks are identified by µPL. The peak at 0.96 eV is the ground state (GS). The ones at 1.05 and 1.12 eV that increased with the optical power density are the first (1ES) and second (2ES) excited states, respectively. The PL intensity of InAs QDs grown on GaAs/Ge/Si is lower than the reference one on a GaAs substrate (not shown here) by a factor of 4. This could be explained by the remaining threading dislocations in the GaAs/Ge/Si structure acting as non-radiative recombination centers [16]. PL peak’s Full Width at Half Maxima (FWHM), around 70 meV, are very close for both samples.

To evaluate the mean shape and size of a large number of buried QDs contributing to the observed PL band, numerical modeling, combined with PL spectroscopy can be efficiently employed [17]. Indeed, InAs QDs ground and excited states emission energies can be numerically determined by solving the single-band 3D Schrödinger equation in Cartesian coordinates using the finite elements method within the effective mass approximation [18,19]. The Schrödinger equation, given by Equation (1), is solved for electrons and heavy holes.
(1)$$-\frac{\hbar^{2}}{2}\nabla \left(\frac{1}{m^{*}\left(\vec{r}\right)}\nabla \emptyset \left(\vec{r}\right)\right)+V\left(\vec{r}\right)\emptyset \left(\vec{r}\right)=E\emptyset \left(\vec{r}\right)$$
m*, E, and ∅ are the carrier’s effective mass, confined energy levels, and wave function, respectively. V is the confining potential barrier considering the lattice mismatch induced stain effects [17] and r is the Cartesian coordinates’ vector.

InAs and GaAs parameters were taken from [20]. Calculations were performed assuming the QDs have a truncated pyramidal shape, in line with TEM observations (Figure 4). Furthermore, the In/Ga intermixing at the QDs/barrier interface was assumed to be negligible, enabling us to consider QDs to be made of pure InAs in our calculations. Indeed, stabilized surfaces with bonded InAs are favorable in MOCVD, preventing indium segregation and intermixing at temperatures higher than those typically employed in solid source molecular beam epitaxy [21,22]. Furthermore, the low InAs deposition rate results in bigger dots which are less sensitive to interface intermixing [23]. Additionally, the presence of InGaAs underlying layer may also have minimized the intermixing at the QD bottom [24].

Inter-band transition energies were numerically tuned by changing the truncated pyramid height, base length, and ratio between upper and base surfaces. The matching of the theoretical emission energies with the PL emission peak’s maxima yielded an accurate estimation of the buried QD size and shape. Figure 6 shows a schematic of the modeled QD.

Experimental and theoretical emission energies associated with the ground states and the first and second excited states of InAs QDs at the center, mid radius and close to the edges of the 300 mm wafer are provided in Figure 7. Data show the best matching from simulations.

QDs are found to have a constant base side around 29 nm regardless of their position on the wafer. However, the mean dots’ height decreases, from 4.8 nm at the wafer center to 4.6 nm at mid radius, together with a small increase of the top to base surface ratio, from 20% to 25%. However, at the wafer’s edges, the QD height and top to base surface ratio were found to be 3.8 nm and 35%, respectively. The decrease of the QD size at the wafer edge is consistent with the small dot’s density difference evidenced by AFM. In other words, QDs size and density changes are rather small over the surface of the wafer. Following the same procedure for the reference InAs QDs grown on GaAs substrate also yielded truncated pyramidal shaped dots, with base lengths of 28.5 nm and heights of 4.1 nm.

QDs’ dimensions from simulations are slightly lower than those from cross-sectional TEM. They are nevertheless in reasonable agreement. Simulated values show the good uniformity, over the wafer surface, of the QDs dimensions, and the limited impact of intermixing at the interfaces. Further characterization by HR-TEM would nevertheless be required for confirmation. Three-period {InAs QDs/GaAs caps} superlattices were grown on GaAs/Ge/Si(001) instead of a single bi-layer, to be closer to actual devices, where high efficiency emission in the active region is required. Room temperature µPL measurements were carried out automatically on 673 different positions to map the entire 300 mm wafer. We analyzed the three main peaks corresponding to the ground state, GS, and the two excited states, 1ES and 2ES. For clarity, we chose to present their: (i) wavelength position; (ii) intensity; and (iii) FWHM maps (Figure 8).

This figure shows that the µPL signal is uniform over the wafer for the three-period InAs QDs. The energies are centered at 1.000 ± 0.006 (GS), 1.040 ± 0.005 (1ES), and 1.090 ± 0.004 eV (2ES). Their full widths at half maximums (FWHM) are 54 ± 5, 46 ± 5, and 66 ± 7 meV over the 300 mm wafer. Clearly, this cartography confirms the homogeneity of three-period InAs QDs, which is mandatory for O-band emitting light source fabrication in an CMOS industrial environment.

## 4. Conclusions

O-band emitting InAs/GaAs quantum dots were grown in an industrial MOCVD reactor on 300 mm diameter Ge-buffered Si (001) substrates. Morphological investigation by AFM showed a good uniformity of the dots size and distribution over the 300 mm wafer surface. TEM, PL spectroscopy, and numerical modeling revealed that InAs QDs capped with GaAs had a truncated pyramidal shape with a mean height around 4.8 nm and a base length of 29 nm. InAs QDs on Ge/Si substrates had properties similar to those of QDs on GaAs substrates. A slight size increase and a reduced luminescence intensity, most likely due to the higher TDD, were evidenced. Our results demonstrate that InAs/GaAs QDs can be successfully grown on 300 mm Ge/Si substrates, paving the way towards the low-cost integration of O-band emitting light sources on Si(001) wafers.

## Figures and Tables

**Figure 1 nanomaterials-10-02450-f001:**
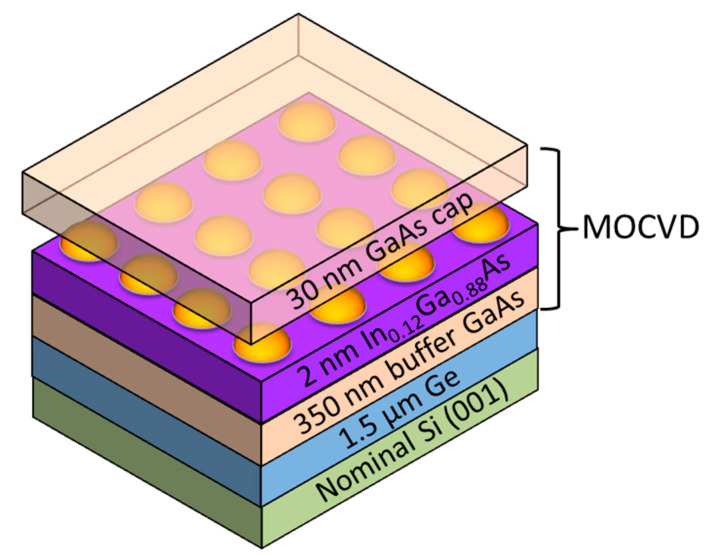
InAs/GaAs quantum dot structure grown on Ge on quasi-nominal Silicon(001) substrates.

**Figure 2 nanomaterials-10-02450-f002:**
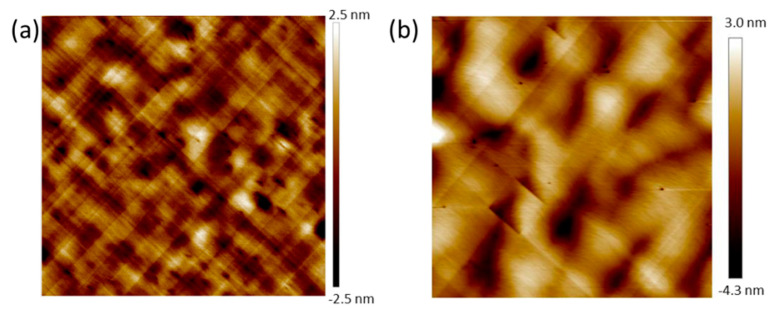
(**a**) A 20 µm × 20 µm AFM image of the surface of a 1.5 µm thick Ge SRB on a Si(001) substrate; and (**b**) a 5 µm × 5 µm AFM image of the 350 nm thick GaAs buffer on a Ge-buffered silicon (001) substrate.

**Figure 3 nanomaterials-10-02450-f003:**

The 1 µm × 1 µm AFM images of uncapped InAs/GaAs QDs grown on a Ge/Si 300mm substrate: (**a**) the center; (**b**) the mid-radius; and (**c**) close to the edge.

**Figure 4 nanomaterials-10-02450-f004:**
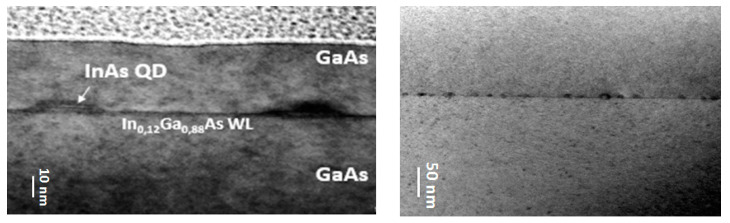
Cross-sectional TEM images of InAs/GaAs quantum dots grown on Ge/Si 300 mm quasi-nominal substrates.

**Figure 5 nanomaterials-10-02450-f005:**
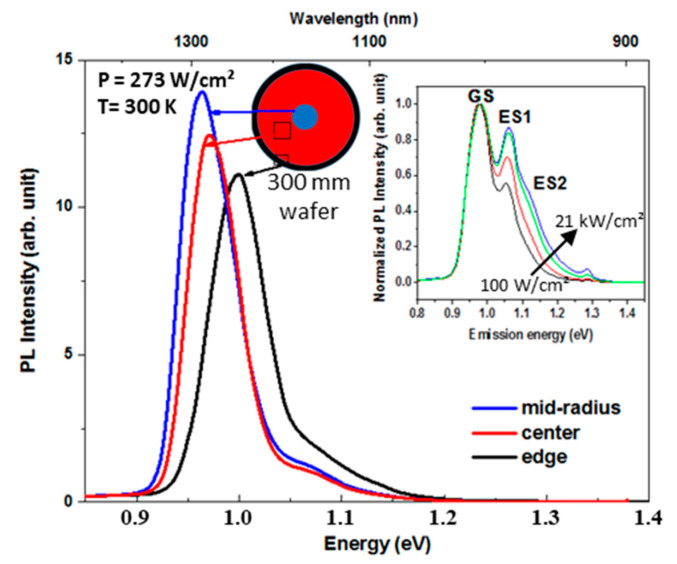
Room temperature PL spectra from InAs QDs grown on GaAs/Ge/Si structure at the center, at the mid-radius and the edge of the wafer with a power density of 273 W/cm^2^. The inset shows the evolution of the normalized µPL spectra as function of the increased optical power density from 100 W/cm^2^ (black line) to 21 kW/cm^2^ (blue line).

**Figure 6 nanomaterials-10-02450-f006:**
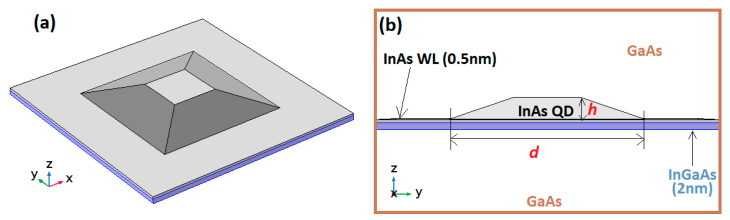
Schematics of the modeled QD structure: (**a**) 3D view; and (**b**) cross-sectional view showing the layers thicknesses and QD dimensions. h and d are the QDs’ height and base side length, respectively.

**Figure 7 nanomaterials-10-02450-f007:**
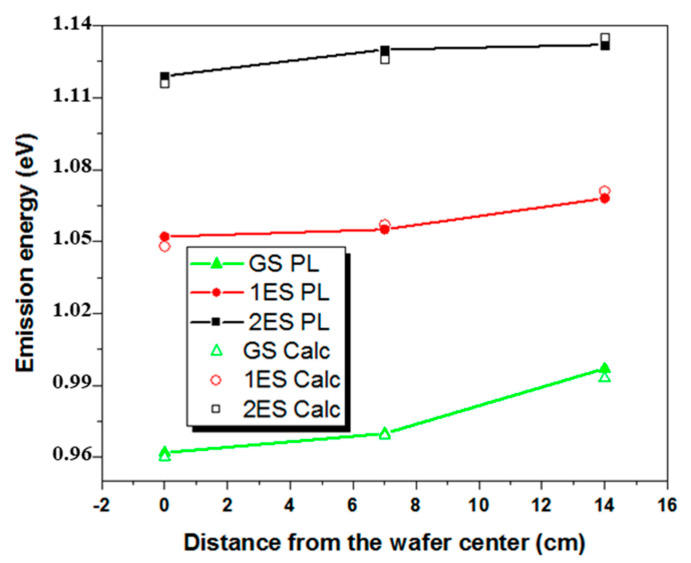
Experimental and calculated emission energies from the QDs’ Ground states (GS) (black squares) and first (1ES) (red circles) and second excited states (2ES) (green triangles) used to determine the mean dots size and shape: lines and full symbols are experimental PL data while empty symbols show the numerical values. The error made on the determination of the peak position is estimated to be less than 1%.

**Figure 8 nanomaterials-10-02450-f008:**
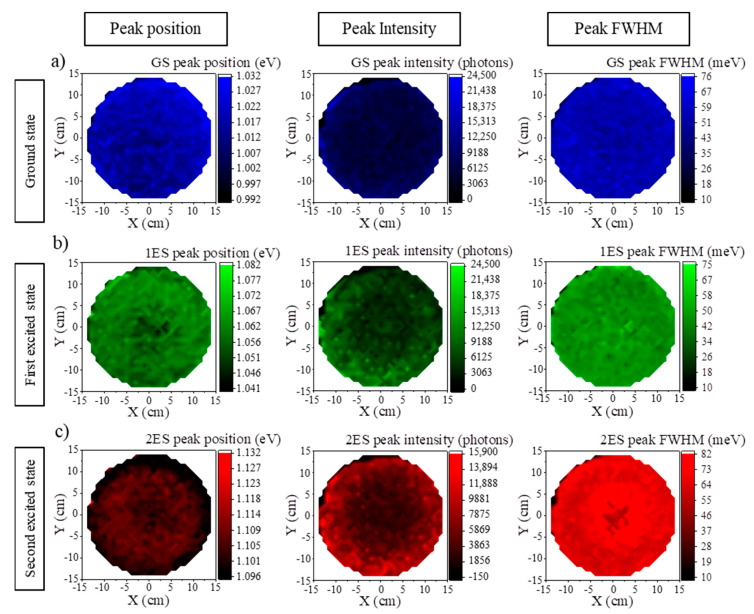
Room temperature µPL 300 mm mapping of the energy, intensity, and FWHM for three stacked InAs QDs grown on GaAs/Ge/Si (001) structure for: (**a**) GS peak; (**b**) 1ES peak; and (**c**) 2ES peak. Spectra are acquired at 673 different positions to map entirely the Si 300 mm wafer.

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
