# Peer review of "O-Band Emitting InAs Quantum Dots Grown by MOCVD on a 300 mm Ge-Buffered Si (001) Substrate"

_nanomaterials, 2020, doi:10.3390/nano10122450_

Round 1

Reviewer 1 Report

Authors report about the successful growth of light-emitting structures with InAs QDs on 300 mm quasi-nominal Si(001) wafers with Ge relaxed buffer using industrial MOCVD equipment. The structures demonstrated rather well uniformity of their luminescent properties throughout the whole 300 mm wafers. The results of the article are good from an engineering point of view. However, the question arises about the scientific novelty of the obtained results. It seems that the approaches applied in the article were used earlier, as also evidenced by the references to the relevant papers in the article. In addition, there are several other comments to the article.

  1. It follows from the caption to Figure 5 that the blue PL spectra in the main figure and in the inset were measured under the same conditions. But it seems that the blue spectrum in the inset cannot be obtained by simple normalization to the maximum intensity of the blue spectrum from the main figure.
  2. As indicated in lines 130-131, an agreement between the calculated and measured position of the PL signal from the QDs is achieved by varying the size of the islands. But in the paper the residual elastic strain in QDs is not considered. Besides, the assumption that the QDs are composed of pure InAs (line 129) seems to be unrealistic because of the well-known intermixing process (see, for example, ref. 15 in paper)
  3. Line 65. It seems that references 12, 13 are wrong.
  4. Lines 73-75. One of the main technological advantages of the article is usage of industrial MOCVD equipment to form InAs QDs instead of more expensive MBE growth method. But the growth conditions for QD growth are not pointed out. What was the nominally deposited amount of In? What was the growth rate?
  5. One of the crucial points for industry-relevant growth approach is the obtained wafer-scale uniformity and authors devote a large part of the article (including 3 figures) to this issue. The authors claimed that they achieved rather high uniformity of structural and optical properties along 300 mm wafer. However they did not explain how such a high uniformity was obtained. Is it just a merit of the used equipment or some special technological approaches were implemented?

Author Response

Dear Editor,

Hereby we resubmit a revised version of our manuscript “O-Band Emitting InAs Quantum Dots Grown By MOCVD On A 300 mm Ge/Si 001 Substrate”.

We appreciated very much the positive feedback of the two referees who judged our work interesting.

Thanks to their suggestions, we have improved the presentation of the work and lifted the critical points spotted by the referees.

The attached letter answers their comments in detail and explains the changes to the manuscript.

Best regards

Referee 1:

Comments and Suggestions for Authors

Authors report about the successful growth of light-emitting structures with InAs QDs on 300 mm quasi-nominal Si (001) wafers with Ge relaxed buffer using industrial MOCVD equipment. The structures demonstrated rather well uniformity of their luminescent properties throughout the whole 300 mm wafers. The results of the article are good from an engineering point of view. However, the question arises about the scientific novelty of the obtained results. It seems that the approaches applied in the article were used earlier, as also evidenced by the references to the relevant papers in the article. In addition, there are several other comments to the article.

1- It follows from the caption to Figure 5 that the blue PL spectra in the main figure and in the inset were measured under the same conditions. But it seems that the blue spectrum in the inset cannot be obtained by simple normalization to the maximum intensity of the blue spectrum from the main figure.

response: Authors are completely agree with this comment and we have clarified this point in figure caption. “Room temperature PL spectra from InAs QDs grown on GaAs/Ge/Si structure at the center, at the mid-radius and the edge of the wafer with a power density of 273 W/cm². The inset shows the evolution of the normalized µPL spectra as function of the increased optical power density from  100 W/cm² (black line) to 21 kW/cm² (blue line).”. We have add also this sentence in the text: “The µPL makes it possible to achieve higher power densities for the same nominal laser power (around 4 µm spot size in µPL against around 200 µm in PL).”

2- As indicated in lines 130-131, an agreement between the calculated and measured position of the PL signal from the QDs is achieved by varying the size of the islands. But in the paper the residual elastic strain in QDs is not considered. We have considered the strain effect in our calculation, mentioned in the description of the carriers confining potential. The calculation details can be found in ref 17

Response: For further clarification, we have modified the description of the confining potential as follow:

“V is the confining potential barrier considering the lattice mismatch induced stain effects [17].”

Besides, the assumption that the QDs are composed of pure InAs (line 129) seems to be unrealistic because of the well-known intermixing process (see, for example, ref. 15 in paper)

We agree with the referee on the fact that the intermixing at the InAs/GaAs interface is inevitable. However, such a phenomenon is strongly dependent on the employed growth technique and deposition parameters such as InAs deposition rate and capping conditions and therefore, cannot be accurately quantified by a universal model.

Furthermore, the InAs QDs in the present work have been grown by MOCVD such a technique give rise to more stabilized surfaces with bonded InAs allowing to prohibit the In segregation, reevaporating and intermixing for relatively higher growth temperature than UHV solid source molecular beam epitaxy. Furthermore, the atomic interface intermixing is further limited by the low InAs deposition rate and the presence of InGaAs underlayer. For this reason, we have chosen to consider pure InAs material QDs in our calculation. Which gave reasonable agreement with TEM observation testifying the limited impact of the intermixing in the present case.

The two following texts have been added to the revised version with supporting references (page 5 and 6) to clarify this point

“Furthermore, we assume the In/Ga intermixing at the QDs/barrier interface to be negligible allowing to consider pure InAs material QDs in our calculations. Indeed, stabilized surfaces with bonded InAs is favorized in MOCVD prohibiting the indium segregation and intermixing at relatively higher growth temperature than conventionally employed in solid source molecular beam epitaxy [21-22]. Furthermore, the low InAs deposition rate leads to bigger dots size which are less sensitive to the interface intermixing [23]. Additionally, the presence of InGaAs underlying layer may also compensate the intermixing at the QD bottom [24]. “

“The overall simulation driven QDs sizes, yields slightly smaller values than those observed by cross-sectional TEM while stay in reasonable agreement. The simulated values remain representative of the overall position dependent QDs size and shape evolution trend and testify a limited impact of the interface intermixing in the present case. Further characterization by HR-TEM will be required for confirmation”

3- Line 65. It seems that references 12, 13 are wrong.

Response: The references have been corrected.

4- Lines 73-75. One of the main technological advantages of the article is usage of industrial MOCVD equipment to form InAs QDs instead of more expensive MBE growth method. But the growth conditions for QD growth are not pointed out. What was the nominally deposited amount of In? What was the growth rate?

Response: We are agree for this comment. For more details, we have add this entence in the paper : “Then, InAs QDs were grown at a temperature of 520°C for 40 s. The growth was then interrupted for 20 s under TBAs. Their growth rate was estimated at 0.18 Å/s, with no in-situ control of the deposited indium amount . The QDs were layed on top of a 2 nm thick In0.12Ga0.88As underlying layer”

5- One of the crucial points for industry-relevant growth approach is the obtained wafer-scale uniformity and authors devote a large part of the article (including 3 figures) to this issue. The authors claimed that they achieved rather high uniformity of structural and optical properties along 300 mm wafer. However they did not explain how such a high uniformity was obtained. Is it just a merit of the used equipment or some special technological approaches were implemented?

Response: For the growth, we have used a 300 mm reactor design by Applied Materials which is the leader in the (Si,Ge) epitaxy by MOCVD in CMOS industry. The reactor is very uniform in a specific range of process parameters, especially, the growth temperature (400-700°C) and the pressure (5-20 Torr). We are in these ranges for the studies presented in the paper.

Reviewer 2 Report

In the manuscript an interesting approach to integrate III-V semiconductor quantum dots (QDs) on Si substrate is presented. It is achieved by growing an additional Ge intermediate buffer layer. This approach results in a significant reduction of threading dislocation density by two orders of magnitude. Moreover, consistent optical, AFM and TEM measurements of these QDs are presented. The manuscript is clearly written and the results worth publishing. Only a few minor points should be addressed before publishing:

  1. What is the reason to use micro-PL instead of a standard PL measurement?
  2. Why any sharp emission lines from individual dots cannot be distinguished in the micro-PL spectra? The QDs density does not look particularly large based on AFM.
  3. I’m confused by Figure 5. The PL spectrum presented in this figure should correspond to the blue spectrum within the inset. But, it does not (?). Is log-scale used in the inset?
  4. On one hand, it is written that QDs density is rather uniform, page 3, line 91. On the other hand, it is stated that “The decrease of the QD size at the wafer edge is consistent with the reduced surface density observed by AFM“,  on page 6, line 148. These two statements are not consistent with each other.

I understand that the changes of the QDs density and QD sizes are rather small depending on the position on the wafer, but it should be written more clearly on page 6, in my opinion.

  1. It is assumed that the QD are made of pure InAs for calculations? What about Ga/In intermixing at the interface? A discussion of this point could be added.

Author Response

Dear Editor,

Hereby we resubmit a revised version of our manuscript “O-Band Emitting InAs Quantum Dots Grown By MOCVD On A 300 mm Ge/Si 001 Substrate”.

We appreciated very much the positive feedback of the two referees who judged our work interesting.

Thanks to their suggestions, we have improved the presentation of the work and lifted the critical points spotted by the referees.

The attached letter answers their comments in detail and explains the changes to the manuscript.

Best regards

Referee 2:

Comments and Suggestions for Authors

In the manuscript an interesting approach to integrate III-V semiconductor quantum dots (QDs) on Si substrate is presented. It is achieved by growing an additional Ge intermediate buffer layer. This approach results in a significant reduction of threading dislocation density by two orders of magnitude. Moreover, consistent optical, AFM and TEM measurements of these QDs are presented. The manuscript is clearly written and the results worth publishing. Only a few minor points should be addressed before publishing:

1- What is the reason to use micro-PL instead of a standard PL measurement?

Response: For more clarity, we have modified this paragraph.

“Photoluminescence measurements were performed at 300 K using a 532 nm diode-pumped solid-state laser with an excitation power density of 273 W/cm2. Micro-photoluminescence measurements were performed at 300 K using a 780 nm diode-pumped solid-state laser with an excitation power density between  100 W/cm2 and 21 kW/cm2.Then, the PL emission was dispersed by a spectrometer and detected by liquid nitrogen cooled InGaAs (IGA) photodetector. The µPL makes it possible to achieve higher power densities for the same nominal laser power (around 4 µm spot size in µPL against around 200 µm in PL).”

2- Why any sharp emission lines from individual dots cannot be distinguished in the micro-PL spectra? The QDs density does not look particularly large based on AFM.

Response: Sharp emission lines could not be observed in our case for the following raisons: The emission peak linewidth of single InAs/GaAs QDs lies between 10 and 15 meV at room temperature. Additionally the laser spot size in microPL is around 4 um exciting simultaneously tens of QDs.

3- I’m confused by Figure 5. The PL spectrum presented in this figure should correspond to the blue spectrum within the inset. But, it does not (?). Is log-scale used in the inset?

Response: Authors are completely agree with this comment and we have clarified this point in figure caption. “Room temperature PL spectra from InAs QDs grown on GaAs/Ge/Si structure at the center, at the mid-radius and the edge of the wafer with a power density of 273 W/cm². The inset shows the evolution of the normalized µPL spectra as function of the increased optical power density from  100 W/cm² (black line) to 21 kW/cm² (blue line).”. We have add also this sentence in the text: “The µPL makes it possible to achieve higher power densities for the same nominal laser power (around 4 µm spot size in µPL against around 200 µm in PL).”

4- On one hand, it is written that QDs density is rather uniform, page 3, line 91. On the other hand, it is stated that “The decrease of the QD size at the wafer edge is consistent with the reduced surface density observed by AFM“,  on page 6, line 148. These two statements are not consistent with each other. I understand that the changes of the QDs density and QD sizes are rather small depending on the position on the wafer, but it should be written more clearly on page 6, in my opinion.

Response: We agree with the referee comment and we have modify this sentence.

“The decrease of the QD size at the wafer edge is consistent with the slight difference observed in the dot’s density by AFM. In other words, the changes of the QDs sizes and density are rather small depending on the position on the wafer.”

5- It is assumed that the QD are made of pure InAs for calculations? What about Ga/In intermixing at the interface? A discussion of this point could be added.

Response: We thank the reviewer for point this out. Indeed, the InAs QDs in the present work have been grown by MOCVD such a technique give rise to more stabilized surfaces with bonded InAs allowing to prohibit the In segregation, reevaporating and intermixing for relatively higher growth temperature than UHV solid source molecular beam epitaxy. Furthermore, the atomic interface intermixing is further limited by the low InAs deposition rate and the presence of InGaAs underlayer. For this reason, we have chosen to consider pure InAs material QDs in our calculation. Which gave reasonable agreement with TEM observation testifying the limited impact of the intermixing in the present case.

The two following texts have been added to the revised version with supporting references (page 5 and 6) to clarify this point

“Furthermore, we assume the In/Ga intermixing at the QDs/barrier interface to be negligible allowing to consider pure InAs material QDs in our calculations. Indeed, stabilized surfaces with bonded InAs is favorized in MOCVD prohibiting the indium segregation and intermixing at relatively higher growth temperature than conventionally employed in solid source molecular beam epitaxy [21-22]. Furthermore, the low InAs deposition rate leads to bigger dots size which are less sensitive to the interface intermixing [23]. Additionally, the presence of InGaAs underlying layer may also compensate the intermixing at the QD bottom [24]. “

“The overall simulation driven QDs sizes, yields slightly smaller values than those observed by cross-sectional TEM while stay in reasonable agreement. The simulated values remain representative of the overall position dependent QDs size and shape evolution trend and testify a limited impact of the interface intermixing in the present case. Further characterization by HR-TEM will be required for confirmation.”

Reviewer 3 Report

nanomaterials-993049-V2

O-band emitting InAs quantum dots grown by MOCVD on a 300 mm Ge/Si 001 substrate”

Comments and recommendations

In the proposed manuscript the authors present a study on InAs quantum dots epitaxial growth by MOCVD on 300 mm Ge-buffered Si (001) substrates. The devices are characterized in AFM, PL and µPL and compared with devices grown on GaAS substrates. In addition, the emission energies are compared with simulated model showing dependency on the geometrical dimensions.

In my opinion the manuscript is nice presented and have sufficient level of novelty for publication. On the other hand, it needs some minor corrections and clarifications prior of publication:

Title:

In my opinion the term “300 mm Ge-buffered Si (001) substrates” is more suitable than “300 mm Ge/Si 001 substrates”

Abstract:

  1. Page 1 line 18: “The hetero-epitaxy of III-V semiconductors…” in my opinion such statement belongs to the introduction and not to the abstract.
  2. Page 1 line 21: “…Ge/Si (100)…” or Ge/Si 001? Stay consistent with the nomenclature.
  3. Page 1 line 23: 1) “…entire 300 mm Si substrate…” is it Si substrate or Ge/Si substrate? Stay consistent. Principally, it should be stated that “The wafer level µPL mapping shows….”
    2) “…shows the homogeneity…emitting at 1.3 µm…” please add statistical values 1.3 ± x.x µm etc.
  4. If I understand the abstract correctly, the authors compare the same QD devices on two different substrates (Ge on Si and GaAs). Here my suggestion is to re-write the abstract as the manuscript “story”. It is important to have a swift message to the reader; what is done and what are the main achievements.

Introduction:

Well written.

Experiment:

Well described.

Results and discussion:

  1. Page 3 line 97: ”The QDs density is rather uniform… the biggest ones” please pay attention that experimental measured values should be displayed with measurement error or standard distribution value. For example, the height is 5.0 ± x.x nm etc.
  2. 5: the figure needs some “cosmetic” modifications: 1) the diagram of the wafers locations needs small title like “300 mm wafer”. One may have the impression that this circular diagram is a single QD. 2) the inset power density levels are not clearly marked.
  3. Page 5 line 149: “Experimental and theoretical emission energies… are provided in Figure 7. 1)how the theoretical emission energies is calculated / dependent on the location on the wafer? Please give short explanation. 2) please add to fig. 7 error bars.
  4. 8 it is not so clear what is the difference between the three wafers (columns) please explain and indicate in the figure.
  5. Page 7 line 181: please pay attention for the meaningful decimal accuracy 1.00 ± 0.01 eV or 1.000 ± 0.006 eV but not 1.00 ± 0.006 etc.

Reference:

fine.

Author Response

Dear Editor,

Hereby we resubmit a second revised version of our manuscript “O-Band Emitting InAs Quantum Dots Grown By MOCVD On A 300 mm Ge-buffered Si (001) Substrate”.

We appreciated very much the positive feedback of the referee.

The attached letter answers their comments in detail and explains the changes to the manuscript. In the revised manuscript, modifications are marked in blue.

Best regards

Reviewer #3

Comments and recommendations

In the proposed manuscript the authors present a study on InAs quantum dots epitaxial growth by MOCVD on 300 mm Ge-buffered Si (001) substrates. The devices are characterized in AFM, PL and µPL and compared with devices grown on GaAs substrates. In addition, the emission energies are compared with simulated model showing dependency on the geometrical dimensions.

In my opinion the manuscript is nice presented and have sufficient level of novelty for publication. On the other hand, it needs some minor corrections and clarifications prior of publication:

Title:

In my opinion the term “300 mm Ge-buffered Si (001) substrates” is more suitable than “300 mm Ge/Si 001 substrates”

We are agree with referee that this term is more suitable so we take it into account in our revised manuscript.

“O-Band Emitting InAs Quantum Dots Grown by MOCVD On a 300 mm Ge-buffered Si (001) Substrate.”

Abstract:

  1. Page 1 line 18: “The hetero-epitaxy of III-V semiconductors…” in my opinion such statement belongs to the introduction and not to the abstract.

We have modify this sentence.

  1. Page 1 line 21: “…Ge/Si (100)…” or Ge/Si 001? Stay consistent with the nomenclature.

This is done in the revised manuscript.

  1. Page 1 line 23: 1) “…entire 300 mm Si substrate…” is it Si substrate or Ge/Si substrate? Stay consistent. Principally, it should be stated that “The wafer level µPL mapping shows….”

Thank you for the correction, we modify the sentence in line 23 by this one:

A wafer level µPL mapping of the entire 300 mm Ge/Si substrate shows the homogeneity of the 3-stacked InAs QDs emitting at 1.3 µm at room temperature.

  1. “…shows the homogeneity…emitting at 1.3 µm…” please add statistical values 1.3 ± x.x µm etc.

The statistical values is added in the revised manuscript.

  1. If I understand the abstract correctly, the authors compare the same QD devices on two different substrates (Ge on Si and GaAs). Here my suggestion is to re-write the abstract as the manuscript “story”. It is important to have a swift message to the reader; what is done and what are the main achievement

We precise in line 22 in the abstract that the InAs QDs are grown on Ge/Si(001) and GaAs(001) in line 22:

“on quasi-nominal 300 mm Ge/Si and GaAs (001) substrates”.

Thank you, this makes the “story” of our manuscript more clearly.

Introduction:

Well written

Experiment:

Well described.

Results and discussion

  1. Page 3 line 97: ”The QDs density is rather uniform… the biggest ones” please pay attention that experimental measured values should be displayed with measurement error or standard distribution value. For example, the height is 5.0 ± x.x nm etc.

We estimate the AFM measurement error for the InAs QDs height of 0.11 nm while for the diameter it is higher with a value of 3 nm. These errors are added in line 102 and 103:

The average QDs height is around 5 ± 0.11 nm, while their diameter is close to 40 ± 3 nm for the biggest ones.

  1. Fig. 5: the figure needs some “cosmetic” modifications: 1) the diagram of the wafers locations needs small title like “300 mm wafer”. One may have the impression that this circular diagram is a single QD. 2) the inset power density levels are not clearly marked.

We change Fig. 5 by adding “300 mm Wafer” title to clarify that it is a 300 mm wafer and we add the optical power density evolution by an arrow with the start and the end values as shown in the inset.

  1. Page 5 line 149: “Experimental and theoretical emission energies… are provided in Figure 7. 1)how the theoretical emission energies is calculated / dependent on the location on the wafer? Please give short explanation. 2) please add to fig. 7 error bars.

The theoretical emission energies depend on the size, the shape, the environment of the quantum dots, so the calculation does not depend on the location on the wafer; this is a theoretical value. The error on Fig7 is estimated to be less than 1%, so error bars are too small to be represented. We have added in the legend that the error made on the determination of the peak position is less than 1%.

  1. Fig. 8 it is not so clear what is the difference between the three wafers (columns) please explain and indicate in the figure.

Fig. 8 is change by the one below where we add a title for each column and line as shown in the figure below:

  1. Page 7 line 181: please pay attention for the meaningful decimal accuracy 1.00 ± 0.01 eV or 1.000 ± 0.006 eV but not 1.00 ± 0.006 etc.

Thank you, This is corrected in the revised manuscript as below:

Line 189 : The energies are centered at 1.000 ± 0.006 eV (GS), 1.040 ± 0.005 eV (1ES) and 1.090 ± 0.004 eV (2ES).  

Reference:

fine.

Round 2

Reviewer 1 Report

I’d like to thank the authors for their responses to my comments. But the main remark to the article remained unanswered:

What is the scientific, not engineering, novelty of the manuscript, which would indicate the necessity of its publication in such a highly rated journal as “Nanomaterials”?

In addition, I consider the answer to my comment about ignoring the In/Ga intermixing process in QDs to be unconvincing. The reference â„–23 which is given by the authors only confirms the intermixing process in QDs. According to the results of that paper, at the growth rates used in the reviewed manuscript, the content of Ga in QDs should be more than 20%.

Besides that, I have not found an article with is cited as Ref. 24.

Based on the above mentioned considerations, I believe that this manuscript should not be accepted for publication in the "Nanomaterials" journal.

Author Response

Dear Editor,

Hereby we resubmit a second revised version of our manuscript “O-Band Emitting InAs Quantum Dots Grown By MOCVD On A 300 mm Ge-buffered Si (001) Substrate”.

We appreciated very much the positive feedback of the referees.

Best regards

24. M. Bruls, J. W. A. M. Vugs, P. M. Koenraad, H. W. M. Salemink, J. H. Wolter, M. Hopkinson,  Appl. Phys. Lett., Vol. 81, No. 9, 1708 (2002) https://doi.org/10.1063/1.1504162